# Non-destructive monitoring of annual trunk increments by terrestrial structure from motion photogrammetry

**Martin Mokroš**[1,2]*, **Jozef Výbošťok**[3], **Alžbeta Grznárová**[4], **Michal Bošela**[4], **Vladimír Šebeň**[5], **Ján Merganič**[2]

**1** Faculty of Forestry and Wood Sciences, Czech University of Life Sciences Prague, Prague, Czech Republic, **2** Department of Forest Harvesting, Logistics and Ameliorations, Faculty of Forestry, Technical University in Zvolen, Zvolen, Slovakia, **3** Department of Forest Economics and Management, Faculty of Forestry, Technical University in Zvolen, Zvolen, Slovakia, **4** Department of Forest Resource Planning and Informatics, Faculty of Forestry, Technical University in Zvolen, Zvolen, Slovakia, **5** National Forest Centre–Forest Research Institute Zvolen, Zvolen, Slovak Republic

* mokros@fld.czu.cz

**Data Availability Statement:** All relevant data are within the paper and its Supporting Information files.

## Abstract

Annual trunk increments are essential for short-term analyses of the response of trees to various factors. For instance, based on annual trunk increments, it is possible to develop and calibrate forest growth models. We investigated the possibility of estimating annual trunk increments from the terrestrial structure from motion (SfM) photogrammetry. Obtaining the annual trunk increments of mature trees is challenging due to the relatively small growth of trunks within one year. In our experiment, annual trunk increments were obtained by two conventional methods: measuring tape (perimeter increment) at heights of 0.8, 1.3, and 1.8 m on the trunk and increment borer (diameter increment) at a height of 1.3 m on the trunk. The following tree species were investigated: *Fagus sylvatica* L. (beech), *Quercus petraea* (Matt.) Liebl. (oak), *Picea abies* (L.) H. Karst (spruce), and *Abies alba* Mill (fir). The annual trunk increments ranged from 0.9 cm to 2.4 cm (tape/perimeter) and from 0.7 mm to 3.1 mm (borer/diameter). The data were collected before- and after-vegetation season, besides the data collection increment borer. When the estimated perimeters from the terrestrial SfM photogrammetry were compared to those obtained using the measuring tape, the root mean square error (RMSE) was 0.25–1.33 cm. The relative RMSE did not exceed 1% for all tree species. No statistically significant differences were found between the annual trunk increments obtained using the measuring tape and terrestrial SfM photogrammetry for beech, spruce, and fir. Only in the case of oak, the difference was statistically significant. Furthermore, the correlation coefficient between the annual trunk increments collected using the increment borer and those derived from terrestrial SfM photogrammetry was positive and equal to 0.6501. Terrestrial SfM photogrammetry is a hardware low-demanding technique that provides accurate three-dimensional data that can, based on our results, even detect small temporal tree trunk changes.

**Funding:** This work was supported by Grant No. CZ.02.1.01/0.0/0.0/16_019/0000803 ("Advanced research supporting the forestry and wood-processing sector's adaptation to global change and the 4th industrial revolution") financed by OP RDE, by the Slovak Research and Development Agency through Grant No. APVV-15-0714 and APVV-18-0305 and by the Scientific Grant Agency of the Ministry of Education, Science, Research, and Sport of the Slovak Republic under the grant VEGA 1/0881/17. The funders had no role in study design, data collection and analysis, decision to publish, or preparation of the manuscript.

**Competing interests:** The authors have declared that no competing interests exist.

## Introduction

Tree trunk increments provide essential information for forest management planning and forest modelling. This is particularly important for the development and calibration of growth models (particularly for allometric equations), selecting tree species for logging and protection, estimating cutting cycles, and describing silvicultural treatments [1]. Furthermore, tree trunk increments are often used as an important component of studies examining the response of tree growth to natural variations or anthropogenic changes in the environment, and they can be used to explore the dynamics of a natural forest as well as land use changes [2,3]. Tree increments vary greatly between and within tree species, as well as in relation to age, season, micro-climatic conditions [4], and stand density [5]. Therefore, conducting measurements for different tree species is necessary along the gradient of climate and site conditions.

Thus far, many types of tree trunk diameter and perimeter measuring instruments and methods have been developed. Accuracy, precision, cost, and operational simplicity are the main properties that differentiate them [6]. These methods can initially be divided into two principal categories: destructive and non-destructive. Destructive methods include the use of an increment borer [7] while non-destructive methods use callipers and measuring tapes [8], rubbery rulers [9] and optical dendrometers [6,10].

The rapid development of technologies able to create three-dimensional (3D) point clouds has generated additional data sources for measurements of tree parameters [11]. The primary acquisition methods for obtaining the 3D data including magnetic motion tracking, terrestrial laser scanning (TLS), and terrestrial structure from motion (SfM) photogrammetry [12]. Magnetic motion tracking can precisely reconstruct a tree trunk surface [13]. However, the device must be moved near to the tree trunk by an operator. Therefore, the upper parts of a trunk are difficult to obtain and the whole process is highly time-consuming. TLS is a technique that has been investigated for forestry usage for approximately 20 years [14]. This method can be used to derive tree parameters including the diameter at breast height (DBH), trunk volume, height, and crown parameters, among others [15–17]. Moreover, TLS uses an active sensor in the form of a laser beam. The alternative technique with a passive sensor is SfM photogrammetry, which automatically reconstructs objects based on two-dimensional digital images [18]. Compared to magnetic motion tracking and TLS, the terrestrial SfM photogrammetry method offers a low-cost and less time-consuming solution [19].

Several authors have used terrestrial SfM photogrammetry to reconstruct trees within plots [20–25] while others focused on individual trees [12,19,26–29]. The root mean square error (RMSE) of the DBH estimation is generally more accurate for studies focused on individual trees, where the RMSE is mainly < 1 cm; whereas, for those in which whole plots are reconstructed, the RMSE is a couple of centimetres.

To the best knowledge of the authors, a study is yet to be published focusing on tree trunk increment estimations from image-based point cloud, by the date of the submission of this manuscript. Meanwhile, the possibility of estimating tree trunk increments from TLS-based point clouds has been researched by several authors [30–32]. Mengesha, Hawkins, and Nieuwenhuis [30] focused on the two-year volume increments of a forest stand of Sitka spruce. They found that the volume increment from the TLS estimation was only 6% (4.77 $m^3$ $ha^{-1}$) different from conventional measurements, when only visible trees for TLS were included. Luoma et al. [32] used TLS to estimate the 9-year changes of tree volume and taper. They proved the possibility of detecting the volume increment from TLS-based point clouds within such a period. Both studies focused on tree increments based on the volume as the main attribute. The use of volume to detect increments should have an advantage, while in theory,

multiple perimeters are included in the whole volume, then the random error from estimation can be decreased.

The aim of this study is to explore the potential of terrestrial SfM photogrammetry to estimate annual tree trunk increments of individual trees, based on their perimeters and diameters. Our hypothesis is that terrestrial SfM photogrammetry will not be capable of detecting annual tree trunk increments of mature trees due to the high variability of the estimation error, which will be larger than the size of the annual trunk increments of mature trees. Two conventional methods were used to compare the results from terrestrial SfM photogrammetry: measuring tape (perimeter) and increment borer (diameter). Furthermore, the influence of four tree species and three different heights on the tree trunks were investigated.

## Methods

Data acquisition using the measuring tape and terrestrial SfM photogrammetry were repeated two times during the year of 2017. The first was performed during March (before-vegetation season), while the second was performed during November (after-vegetation season). We followed the same data acquisition procedures in both periods. The trunk core acquisition by the increment borer was conducted in 2018. After the data acquisition images were processed to point clouds, the perimeters of the trunks were estimated at three different height levels. Next, the annual increments were calculated and compared to evaluate the possibility of using terrestrial SfM photogrammetry for annual trunk increment estimation. A diagram of the detailed workflow is shown in Fig 1.

### Study site

The forest stands where the target trees are situated represent mainly *Fagus sylvatica* L. (beech), *Quercus petraea* (Matt.) Liebl. (oak), *Picea abies* (L.) H. Karst (spruce), and *Abies alba* Mill (fir). These tree species were chosen for the research experiment and 10 trees from each species were selected. The positions of the trees within the forest stands are shown in Fig 2. The geographical coordinates of the centre point of each tree species cluster are as follows: beech (48.646389, 19.0425), oak (48.627778, 19.043611), spruce (48.625833, 19.045278), and fir (48.646111, 19.041667). The ages varied from 55 to 80 years. Each tree species was situated in the same forest stand. Additionally, trees of the same species were of the same age. No specific permissions were required for measurements at research plots locations. All research plots were within the University Forest Enterprise of Technical University in Zvolen which are available for research activities and specific permission was not required. Field studies did not involve endangered or protected species.

### Conventional measurements

We used two conventional methods to measure the annual trunk increments. First, we used a measuring tape to measure the trunk perimeters before- and after-vegetation season. The tree trunk perimeters were measured at three height levels of 0.8, 1.3, and 1.8 m. We paired the measurements from after- and before-vegetation season and subtracted them to calculate the annual trunk increments. Second, we used an increment borer to collect the trunk cores at a height of 1.3 m on trunks in four different directions, and from each collected core, the diameter increments for the year 2017 were recorded. The final annual trunk increment was calculated as an average of the four collected increments.

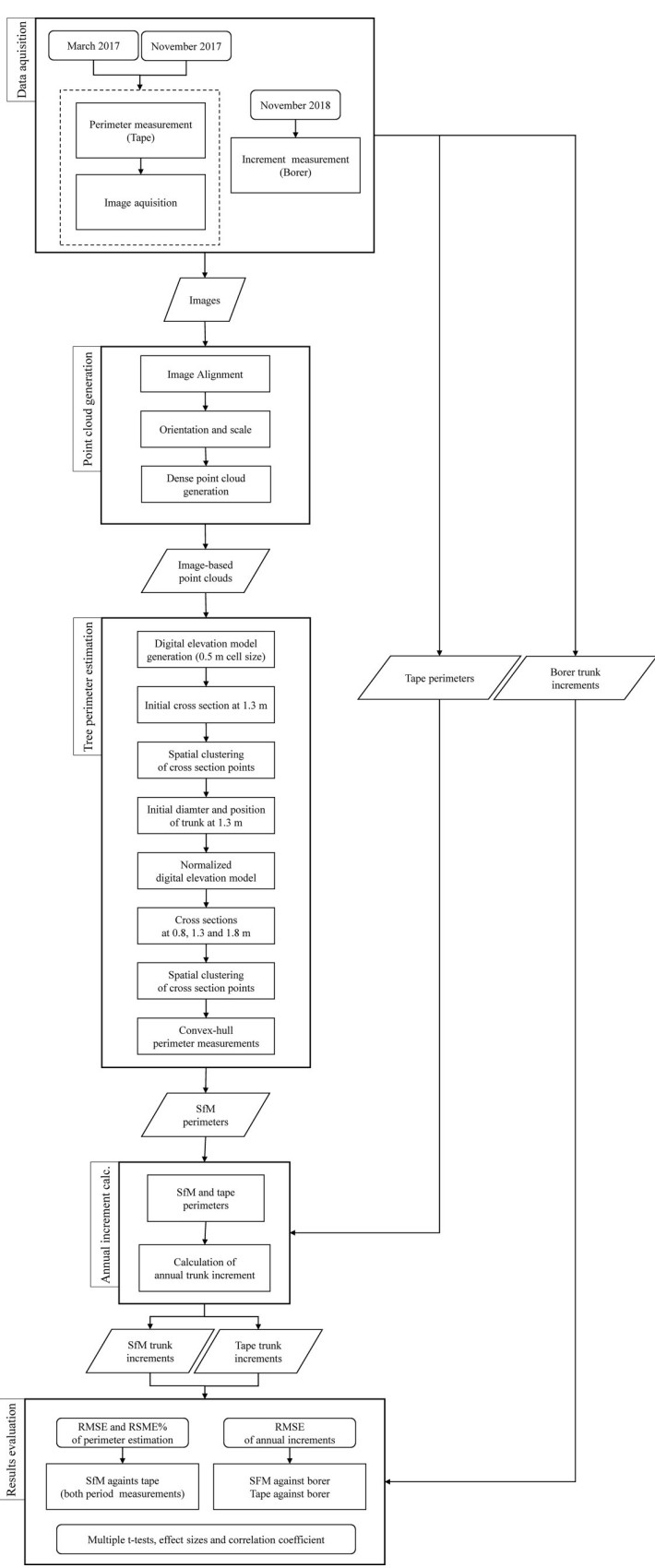

**Fig 1. Diagram of the full workflow.**

## Image acquisition

Ahead of image capturing, we placed 8-bit markers for scaling purposes on the ground; two markers on one paper. Furthermore, an additional marker was placed on the carbon pole to serve the Z-axis orientation. A digital single-lens reflex camera, Canon 70D (Canon Inc., Tokyo, Japan), equipped with a Canon EF 8–15 mm f/4L Fisheye USM, was used to capture the images. The camera has a CMOS sensor and 20.2-megapixel resolution. The lens was fixed to 15 mm. The fisheye lens proved to provide accurate results as well as a shorter acquisition time compared with the non-fisheye lens [19]. A circular-shaped imagery path with a 3-m radius was used. The operator followed this path around a tree and captured images approximately every 0.5 m. Further details on the data acquisition can be found in [19]. The image acquisition was performed in two periods together with the perimeter measurements using the measuring tape (March and November 2017).

## Image-based point cloud generation

Post-processing of images was conducted with the Agisoft Photoscan Professional 1.2.6 software (Agisoft LCC, St. Petersburg, Russia). The images were processed into scaled and oriented point clouds separately within chunks (Fig 3). Each chunk corresponded to one individual tree. The images were aligned with the automatic camera calibration. The alignment settings in which each image is compared to another within the chunk were used in full resolution. The markers were automatically after the alignment. On each piece of A4 paper were two markers of fixed distances. These distances were used to scale the tie point clouds. We began with pairing markers that share the piece of A4 paper and continued with setting a scale. The distance between the centres of the markers was 14.2 cm. The markers placed on the carbon pole were used to set the orientation of the Z-axis. Subsequently, the dense point cloud was generated and exported to .txt format.

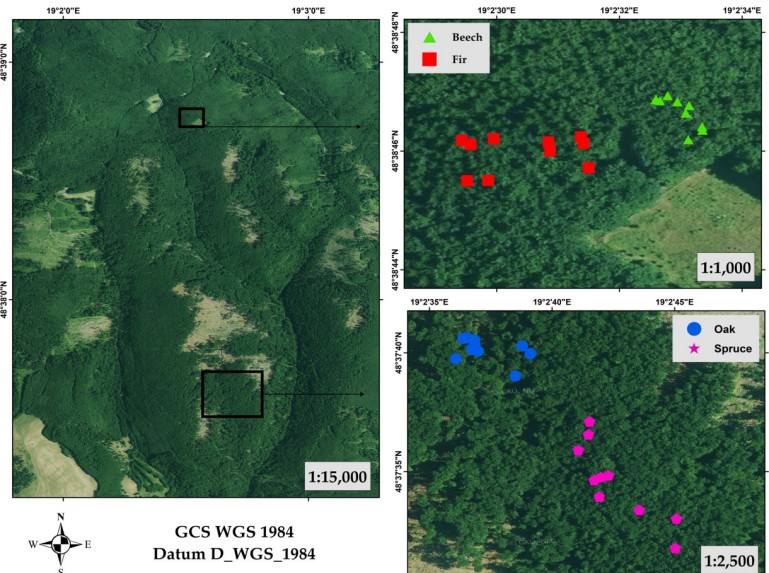

**Fig 2. Study sites with positions of trees within the forest stands.**

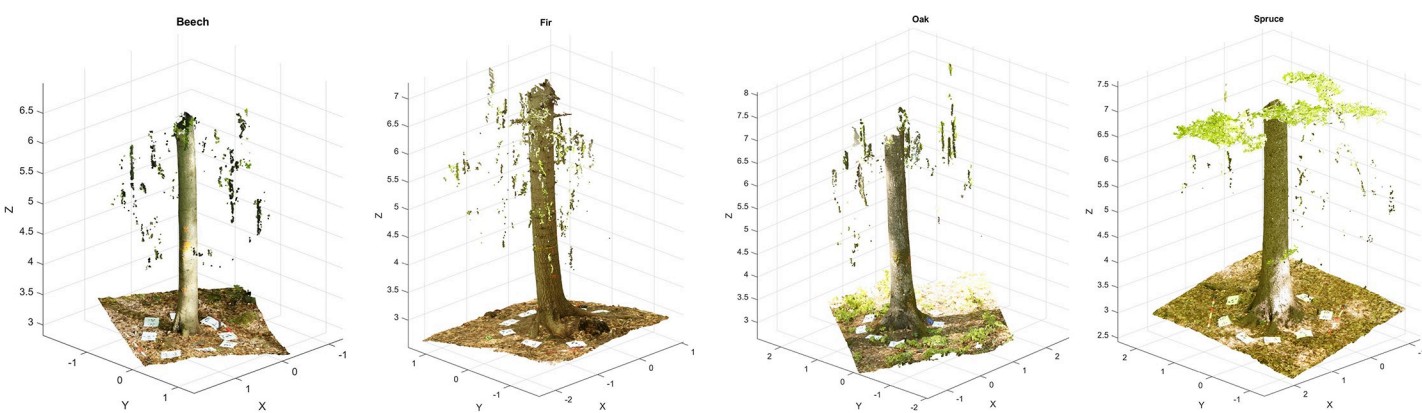

**Fig 3. Examples of dense point clouds of each tree species: Beech, fir, oak, and spruce (starting from the left, respectively).**

## Tree perimeter estimation from image-based point cloud

The cross-sections at 0.8, 1.3, and 1.8 m were created using the DendroCloud software [33,34]. To "cut" the cross-sections at different levels, a digital elevation model (DEM) was generated. The grid size of the DEM was set at 0.5 m. The point with the lowest Z-value was assigned to each cell. Based on the DEM, the initial cross-section at 1.3 m with a 2-cm thickness was generated. The points within the cross-section were spatially grouped to identify the trunk points. To calculate the initial diameter and position of a tree, we used the least-squares fitting of circles algorithm [15]. Based on the obtained position and diameter, we calculated the normalized DEM around the tree and generated multiple cross-sections at the desired heights (0.8, 1.3, and 1.8 m). After, the points at each height were spatially grouped; each tree had three grouped cross-sections. We exported the acquired results into .csv format and then imported them into the ArcGIS for Desktop software (Environmental Systems Research Institute, Redlands, CA, USA). Within ArcGIS for Desktop, the convex hull algorithm was applied by the module Minimum Bounding Geometry to calculate the perimeter. Altogether, 240 perimeters were calculated. Furthermore, all perimeters were divided by $\pi$ to obtain the diameters. Then, the diameters were used to compare the increments with the results from the increment borer. Finally, we paired the estimated perimeters and diameters from after- and before-vegetation season and subtracted them to calculate the annual increment.

## Results evaluation

First, the perimeter estimation error was calculated as the difference between the measurement tape perimeters and estimated perimeters from the terrestrial SfM photogrammetry. We calculated the RMSE and relative RMSE (RMSE%) of the perimeter estimations for each tree species at the three height levels of the measurements.

Furthermore, multiple t-tests were conducted to detect the statistically significant difference between the annual tree increments calculated from the conventional measurements and from the terrestrial SfM photogrammetry.

Furthermore, the effect size was calculated to show the magnitude of the difference between the conventional measurements and estimations from the terrestrial SfM photogrammetry of the perimeters and annual increments. We used Cohen's $d$ effect size [35] and expanded the scale of magnitude: very small-0.01, small-0.20, medium-0.50, large-0.80, very large-1.20, huge-2.0 [36].

**Table 1. Root mean square error (RMSE) (cm) of perimeter estimation from terrestrial SfM photogrammetry.**

|  | Before-vegetation season | | | | | After-vegetation season | | | | |
|---|---|---|---|---|---|---|---|---|---|---|
|  | Beech | Fir | Oak | Spruce | Overall | Beech | Fir | Oak | Spruce | Overall |
| 80 | 0.39 | 0.56 | 0.97 | 0.55 | 0.65 | 0.42 | 1.04 | 1.32 | 1.02 | 1.01 |
| 130 | 0.25 | 0.31 | 1.20 | 0.75 | 0.74 | 0.42 | 1.17 | 0.62 | 0.97 | 0.84 |
| 180 | 0.57 | 0.42 | 1.06 | 0.76 | 0.74 | 0.28 | 0.48 | 0.60 | 1.28 | 0.76 |
| Overall | 0.42 | 0.44 | 1.08 | 0.69 | **0.71** | 0.38 | 0.94 | 0.91 | 1.10 | **0.88** |

Finally, the annual trunk increments obtained from the trunk cores via the measurement tape and the terrestrial SfM photogrammetry were compared. We also compared the average annual trunk increments for each tree species. The RMSE of the annual trunk increments was calculated from the measuring tape and the terrestrial SfM photogrammetry methods toward the annual trunk increments from the trunk cores collected using the increment borer.

## Results

### Perimeter estimation accuracy

First, the RMSE of the perimeter estimations was calculated for both datasets (before- and after-vegetation season). Altogether, 40 trees (10 trees of 4 tree species) were measured and estimated at three different heights. The lowest RMSE was achieved for beech in both data acquisitions and at almost all heights (except before-vegetation season at 1.8-m height). Meanwhile, oak had the highest RMSE in almost all cases. In general, the RMSE varied from 0.25 to 1.32 cm (Table 1).

Different orders of the RMSE% can be seen corresponding to the tree species. This is caused by the different sizes of the average perimeters between tree species. The RMSE% varied through all datasets from 0.24% to 0.91% (Table 2).

The effect sizes between the estimation errors of the before- and after-vegetation season datasets of each tree species were calculated and separated into categories based on [36]. The effect size of spruce was very small (0.019), those of beech and fir were small (0.284 and 0.482), while that of oak was very large (1.240). Furthermore, Fig 4 shows the correlation coefficients together with linear regression lines. It can be seen that beech had the highest correlation between the errors of datasets measured before- and after-vegetation season ($r = 0.5006$).

### Annual perimeter increment estimation

Table 3 shows the perimeter increments calculated from the data obtained by measurement tape and then estimated using the terrestrial SfM photogrammetry; they vary from 0.9 to 2.4 cm and from 0.9 to 2.5 cm, respectively. The annual trunk increments from both methods were compared by the paired t-test and separated according to tree species. There was no significant difference between the tree trunk increments for fir ($p$-value = 0.057), beech ($p$-

**Table 2. Relative RMSE (%) of perimeter estimation.**

|  | Before-vegetation season | | | | After-vegetation season | | | |
|---|---|---|---|---|---|---|---|---|
|  | Beech | Fir | Oak | Spruce | Beech | Fir | Oak | Spruce |
| 80 | 0.45 | 0.42 | 0.63 | 0.32 | 0.47 | 0.76 | 0.85 | 0.58 |
| 130 | 0.30 | 0.24 | 0.84 | 0.47 | 0.49 | 0.91 | 0.42 | 0.61 |
| 180 | 0.71 | 0.35 | 0.77 | 0.50 | 0.34 | 0.39 | 0.43 | 0.84 |
| Overall | 0.51 | 0.35 | 0.75 | 0.43 | 0.44 | 0.73 | 0.62 | 0.67 |

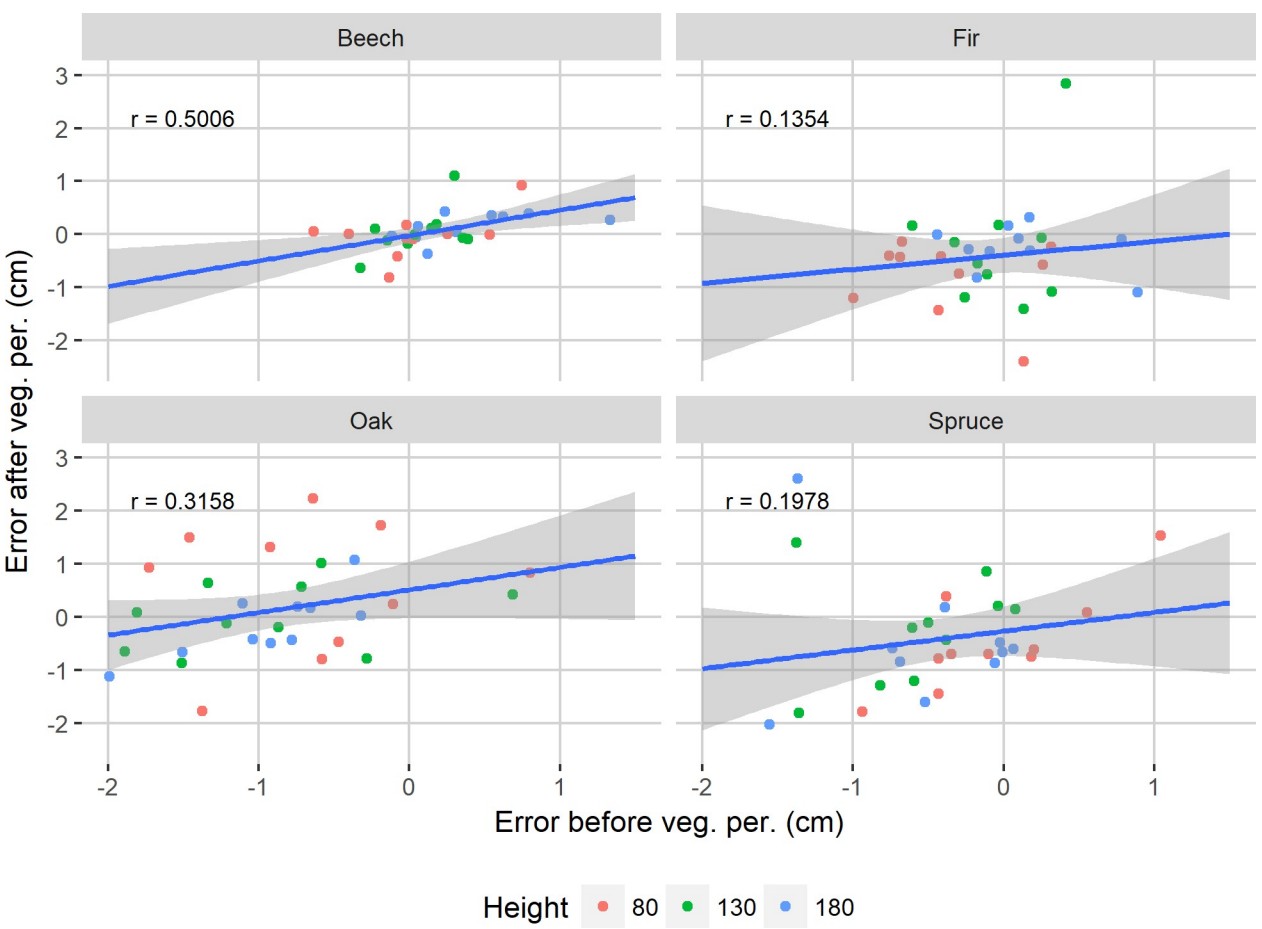

**Fig 4. Scatterplot of estimation errors before- and after-vegetation season.** The correlation coefficients of each tree species are reported and a linear regression line with 95% confidence bands is implemented.

value = 0.130), and spruce ($p$-value = 0.935). Whereas, oak had a statistically significant difference between the annual trunk increments ($p$-value = 0.000003). The detailed results of the t-tests are shown in the S1 Appendix. The effect sizes between the annual trunk increments obtained using the measuring tape and the terrestrial SfM photogrammetry for each tree species are as follows: the effect size of spruce was very small (0.017), those of beech and fir was small (0.290 and 0.401), and that of oak was large (1.152).

The annual increments obtained using the measurement tape and the terrestrial SfM photogrammetry are shown in Fig 5 together with the correlation coefficients. Overall, beech had the strongest correlation ($r$ = 0.5172).

**Table 3. Average annual increment (cm) for tree species separated by height on the tree trunk calculated from conventional measurement data and from terrestrial SfM photogrammetry (estimation).**

| | Measuring tape | | | | Terrestrial SfM photogrammetry | | | |
|---|---|---|---|---|---|---|---|---|
| | **Fir** | **Beech** | **Oak** | **Spruce** | **Fir** | **Beech** | **Oak** | **Spruce** |
| 80 | 2.4 | 1.4 | 1.2 | 1.8 | 2.0 | 1.3 | 2.5 | 1.6 |
| 130 | 1.6 | 1.2 | 1.0 | 1.0 | 1.5 | 1.1 | 2.0 | 1.4 |
| 180 | 1.8 | 1.2 | 0.9 | 1.1 | 1.4 | 0.9 | 1.7 | 1.1 |
| Average | 2.0 | 1.2 | 1.1 | 1.4 | 1.6 | 1.1 | 2.1 | 1.4 |

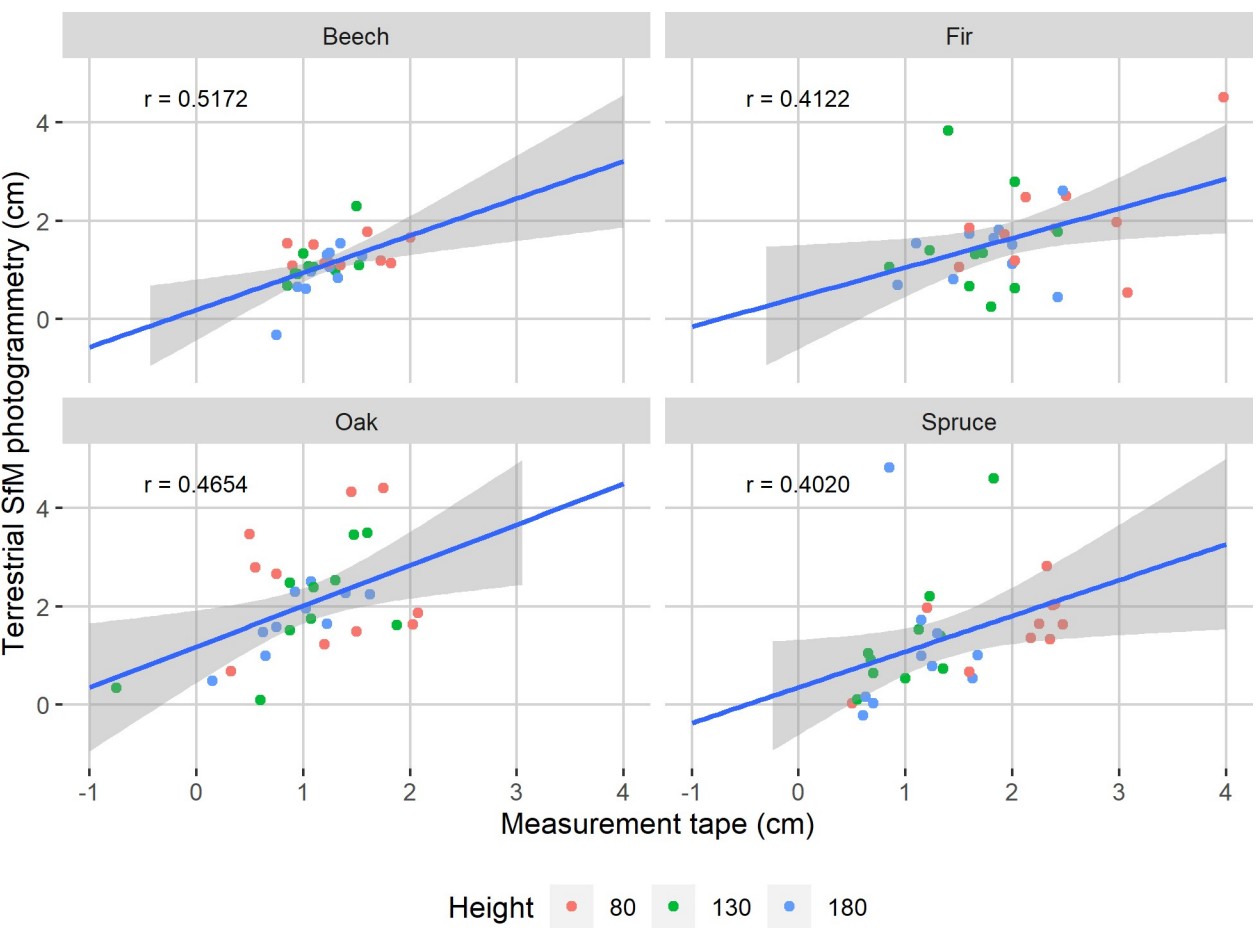

**Fig 5. Scatterplot of perimeter increments calculated from the reference data and from terrestrial SfM photogrammetry.** The correlation coefficients of each tree species are reported and a linear regression line with 95% confidence bands is implemented.

## Core diameter increments

Table 4 shows a comparison of the increments from all three approaches used. Beech had the lowest difference between the conventional methods and the terrestrial SfM photogrammetry. Meanwhile, oak had the highest.

Additionally, we calculated the RMSE of the annual trunk increments obtained from the measuring tape and the terrestrial SfM photogrammetry toward the annual trunk increments obtained from the collected cores by the increment borer. The RMSE varied from 0.4 to 0.9 mm and from 0.7 to 2.1 mm, respectively (Table 5).

The Pearson product-moment correlation coefficient was computed to assess the relationships between the increments obtained from the trunk cores and measuring tape, and the increments from the trunk cores and terrestrial SfM photogrammetry. There was a positive correlation in both cases, $r = 0.7406$ and $r = 0.6501$, respectively (Fig 6).

**Table 4. Comparison of average annual trunk increments (mm) for each tree species calculated from trunk cores, measuring tape, and terrestrial SfM photogrammetry.**

|  | Beech | Fir | Oak | Spruce | Overall |
|---|---|---|---|---|---|
| Borer | 1.6 | 1.9 | 1.6 | 1.2 | 1.6 |
| Tape | 1.8 | 2.7 | 1.9 | 1.7 | 2.0 |
| SfM | 1.8 | 2.4 | 3.1 | 2.2 | 2.4 |

**Table 5. RMSE (mm) of annual trunk increment estimations of measuring tape and terrestrial SfM photogrammetry methods.**

|        | Beech | Fir | Oak | Spruce | Overall |
|--------|-------|-----|-----|--------|---------|
| Tape   | 0.4   | 0.9 | 0.6 | 0.6    | 0.7     |
| SfM    | 0.7   | 1.5 | 2.1 | 1.9    | 1.6     |

## Discussion

Annual trunk increments have been increasingly used across the globe to investigate the growth-climate relationships of trees to advise forest policy when seeking adaptation measures to better prepare for predicting climate change in the future. Terrestrial SfM photogrammetry is a technique that provides the possibility to construct a 3D model of trees with high accuracy and precision. It has the advantages of flexibility and relatively low-cost hardware. However, the question remains of whether terrestrial SfM photogrammetry is capable of detecting annual tree trunk increments. To address this, we established an experiment to investigate the possibility of detecting the annual increments within commonly grown trees at mature ages. The main factor that influenced the estimation accuracy between the tree species was the bark surface. Beech had the lowest RMSE of perimeter estimation and the highest conformity with conventional measurements. Meanwhile, oak had the highest RMSE of perimeter estimation and the highest Cohen effect size. In addition, it was the only tree species with confirmed differences, by the t-test, between the increments derived using the measuring tape and terrestrial SfM photogrammetry. Furthermore, the perimeter estimation accuracy of terrestrial SfM photogrammetry was very high for all tree species; the relative RMSE did not exceed 1% in all cases.

Overall, studies focused on individual tree modelling using terrestrial SfM photogrammetry have achieved high accuracy [12,19]. The accuracy of diameter or perimeter estimations decreases rapidly when the object of the study is a forest stand and multiple trees are reconstructed at once [20,22,37]. In future, to determine the tree increment of multiple trees at once, the possibility of increasing the estimation accuracy of the diameter or perimeter should be investigated.

In this study, a measuring tape was used to measure the reference perimeters and, based on the measurements, the annual tree trunk increment was calculated. The measuring tape has a high accuracy for individual tree trunk perimeter measurements [19,38]. However, an issue remains regarding the accuracy achieved for repeated measurements, particularly for tree trunk increments. An increment borer, used to collect wood cores, was also used. The reason for the use of the trunk cores for the annual trunk increments is based on the assumption that it should produce results nearest the data source to measure the most realistic trunk increments.

To better discuss the accuracy of terrestrial SfM photogrammetry to measure the annually resolved diameter increments revealed by our study, we used an extensive database of tree-ring samples collected within the Slovakian National Forest Inventory [39]. We used this database to quantify the potential variability in radial increments across a wide range of ecological conditions and forest management interventions. The large variability, shown in Fig 7, suggests the great potential of the terrestrial SfM photogrammetry in some parts of forests to be used for quantification of the annually resolved diameter increment. Our assumption is base on the RMSE of perimeter estimation presented. However, a significant part of forests remains unsuitable for measurements using the available terrestrial SfM photogrammetry employed with the current accuracy.

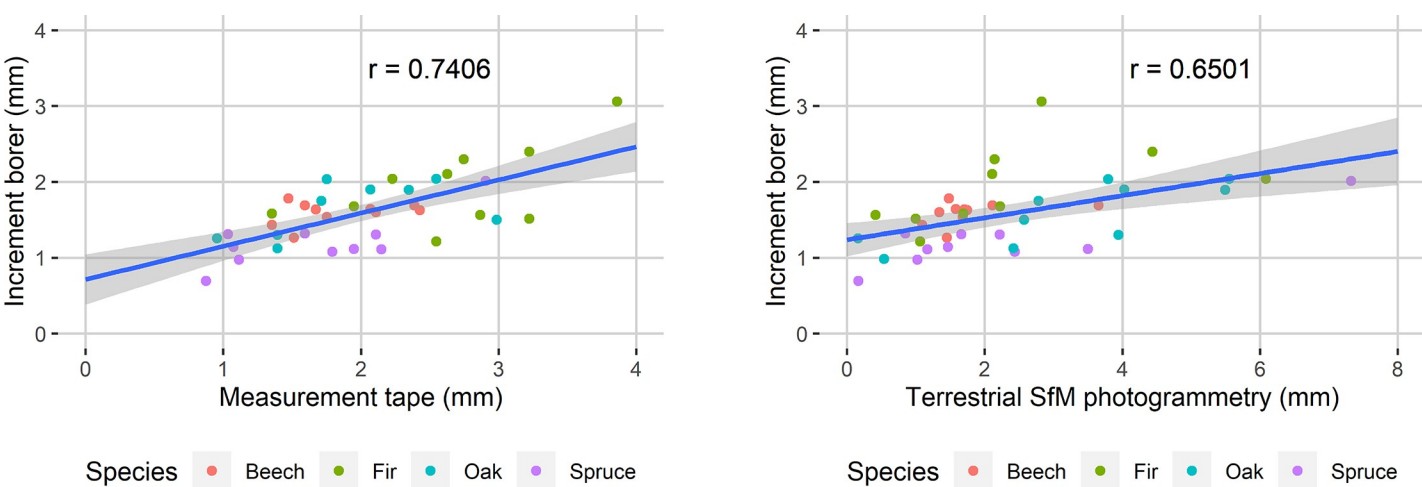

**Fig 6. Scatterplot of annual trunk increments calculated from the measuring tape data and from terrestrial SfM photogrammetry compared to increments collected by the increment borer.** In both, the correlation coefficients are reported and a linear regression line with 95% confidence bands is implemented.

**Fig 7. Diameter increments at breast height measured from tree-ring samples collected from European beech, silver fir, Norway spruce, and Oak sp. trees within the Slovakian National Forest Inventory in 2015–2016.** Line is based on relative RMSE achieved in our research and it is linked to DBH and annual increment.

## Conclusion

In our research we focused on the possibility of estimating annual trunk increments by terrestrial SfM photogrammetry. We found, based on the accuracy and size of the error, that this method is not suitable for small increments. Furthermore, the suitability of this method is even less for tree species with rugged bark, for example oak. Overall, the annual trunk increments for all tree species (European beech, silver fir, Norway spruce, and oak) at all height levels (0.8, 1.3, and 1.8 m) varied from 1.0 to 2.4 cm when measured using the measuring tape. Meanwhile, the RMSE of the annual trunk increment varied from 0.25 to 1.32 cm.

A question also remains regarding the accuracy of all the methods used. The annual trunk increments derived from the measuring tape also introduced some error (compared with the increment borer results), which disqualifies the method from usage in some cases. Additionally, the accuracy of the increment borer method presents another problem. In future research, these doubts should be addressed, and we suggest measuring annual trunk increments by a fully destructive method in which the trunk is harvested and whole cross-sections are analysed to derive the increment around the whole trunk perimeter. Furthermore, the use of terrestrial SfM photogrammetry to estimate the annual trunk increment at the highest levels on a trunk should be investigated.

## Supporting information

**S1 Appendix.**
(DOCX)

## Author Contributions

**Conceptualization:** Martin Mokroš, Jozef Výbošťok, Michal Bošela, Vladimír Šebeň, Ján Merganič.

**Data curation:** Vladimír Šebeň.

**Funding acquisition:** Martin Mokroš, Ján Merganič.

**Methodology:** Martin Mokroš, Jozef Výbošťok, Alžbeta Grznárová.

**Supervision:** Ján Merganič.

**Validation:** Martin Mokroš.

**Visualization:** Martin Mokroš.

**Writing – original draft:** Martin Mokroš, Jozef Výbošťok, Michal Bošela.

**Writing – review & editing:** Martin Mokroš, Jozef Výbošťok, Alžbeta Grznárová, Ján Merganič.

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
