## [Decision Letter · Decision Letter 0]

3 Dec 2019

PONE-D-19-29708

Non-destructive monitoring of annual trunk increments by terrestrial structure from motion photogrammetry

PLOS ONE

Dear Dr. Mokros,

Thank you for submitting your manuscript to PLOS ONE. After careful consideration, we feel that it has merit but does not fully meet PLOS ONE’s publication criteria as it currently stands. Therefore, we invite you to submit a revised version of the manuscript that addresses the points raised during the review process.

Both reviewers require substantial revisions before the manuscript can be properly reviewed. After reviewing the manuscript as well, I agree that the methods require (i) greater detail and (ii) improved writing to clarify existing statements. Indeed, there are cases in the methods description where one would need to guess what has been done. On the bright side, both reviewers believe the work has the potential to be of high interest to a broad readership.

We would appreciate receiving your revised manuscript by Jan 17 2020 11:59PM. To enhance the reproducibility of your results, we recommend that if applicable you deposit your laboratory protocols in protocols.io, where a protocol can be assigned its own identifier (DOI) such that it can be cited independently in the future. For instructions see: http://journals.plos.org/plosone/s/submission-guidelines#loc-laboratory-protocols

We look forward to receiving your revised manuscript.

Kind regards,

John Toland Van Stan II, Ph.D.

Academic Editor

PLOS ONE

Journal Requirements:

2. In your Methods section, please provide additional location information, including geographic coordinates for the data set if available.

Reviewers' comments:

Reviewer's Responses to Questions

**Comments to the Author**

1. Is the manuscript technically sound, and do the data support the conclusions?

Reviewer #1: Yes

Reviewer #2: Yes

2. Has the statistical analysis been performed appropriately and rigorously? 

Reviewer #1: Yes

Reviewer #2: Yes

3. Have the authors made all data underlying the findings in their manuscript fully available?

Reviewer #1: Yes

Reviewer #2: Yes

4. Is the manuscript presented in an intelligible fashion and written in standard English?

Reviewer #1: Yes

Reviewer #2: No

5. Review Comments to the Author

Reviewer #1: General comment

The authors conducted a research on monitoring and accuracy assessment of annual trunk increments by terrestrial structure from motion photogrammetry. First, 240 perimeters of four tree species were measured by tape on three height level (0.8 m, 1.3 m and 1.8 m). Then, these perimeters were estimated by terrestrial structure from motion (SfM) photogrammetry techique. All data monitored from after-and before-vegetation season were used to calculate the annual increment. The paired t-test was used to confirm the statically significant difference between annual tree increments calculated from conventional measurements and from terrestrial SfM photogrammetry. Finally, the accuracy assessment of annual trunk increments determined by tape and terrestrial SfM photogrammetry were compared to results measured by increment borer method. Authors remarked “a significant part of forests remains unsuitable if the available terrestrial SfM photogrammetry were to be employed with current accuracy”.

Here I would like offer some comments as follows

Major comments:

1. Please add a flow chart of experimental steps in Methodology section (consist of image processing, results evaluation and accuracy assessment).

2. In Table 1, could you add mean of perimeter estimated from terrestrial SfM at each height level of four tree species?

3. From line 208 to line 217, the paired t-test was used for evaluating both results, please provide more detail of each tree (t=?, df=?, 95% confidence interval=?).

4. From line 305 to line 309. Could you show clearly a suitable or unsuitable part if the terrestrial SfM photogrammetry were applied in Fig. 6. Also, please make a legend of blue points and red area in Fig.6.

5. Please add conclusion section to reveal that the results archived from terrestrial SfM photogrammetry can or NOT be used for monitoring the annual tree increments based on accuracy assessment. And could the terrestrial SfM photogrammetry technique is replaced by the tape measurement?

Minor comments:

6. Line 260. Term “reconstruct of trees” can be replaced by “construct 3D model of trees”.

7. Line 267. Term “worst RMSE” can be replaced by “highest RMSE”.

8. From line 273 to line 287. Please move to introduction section.

9. I am not an English native speaker but I think the manuscript should be checked by an English native speaker.

Reviewer #2: The paper presents an application of structure from motion photogrammetry for measuring trunk diameter increments. This work is novel as it is the first example of measuring trunk increments using a photogrammetric approach and is likely to be of interest to the readers of Plos One.

The paper requires significant improvement in the writing and general presentation before being published. A number of sections are difficult to interpret. For example, the word distract is used instead of subtract in line 110, and units are missing throughout. Nevertheless, the novelty of the work and methods used appear to be sound as such I recommend minor revisions.

Other comments,

1. Some of the methods are not fully described, for example,

- line 146 "To calculate initial diameter and position of the tree we used the circle fitting algorithm [15]." The circle fitting method is not outlined and reference 15 is a comparison of multiple methods.

- line 153, how was the diameter derived from the polygon?

2. Figure 2 should provide a scale bar for each point cloud

3. The results often repeat the methods

6. PLOS authors have the option to publish the peer review history of their article (what does this mean?). If published, this will include your full peer review and any attached files.

Reviewer #1: Yes: Nguyen Van Trung

Reviewer #2: No

---

## [Author Response · Author response to Decision Letter 0]

21 Jan 2020

Dear John Toland Van Stan II,

Thank you very much for your time and effort. We have answered all questions and comments raised by you and both reviewers.

Editor comment: Both reviewers require substantial revisions before the manuscript can be properly reviewed. After reviewing the manuscript as well, I agree that the methods require (i) greater detail and (ii) improved writing to clarify existing statements. Indeed, there are cases in the methods description where one would need to guess what has been done. On the bright side, both reviewers believe the work has the potential to be of high interest to a broad readership.

Response: We have edited the methods to bring more light to the workflow. Also diagram of whole workflow was added as reviewer 1 suggested. Furthermore we have sent the manuscript to English editing service.

Dear Nguyen Van Trung (Reviewer 1),

thank you very much for your time and effort. Your comments were beneficial and important for our manuscript. In following section, we have answered all your comments and edited our manuscript based on them.

Reviewer #1: General comment

The authors conducted a research on monitoring and accuracy assessment of annual trunk increments by terrestrial structure from motion photogrammetry. First, 240 perimeters of four tree species were measured by tape on three height level (0.8 m, 1.3 m and 1.8 m). Then, these perimeters were estimated by terrestrial structure from motion (SfM) photogrammetry techique. All data monitored from after-and before-vegetation season were used to calculate the annual increment. The paired t-test was used to confirm the statically significant difference between annual tree increments calculated from conventional measurements and from terrestrial SfM photogrammetry. Finally, the accuracy assessment of annual trunk increments determined by tape and terrestrial SfM photogrammetry were compared to results measured by increment borer method. Authors remarked “a significant part of forests remains unsuitable if the available terrestrial SfM photogrammetry were to be employed with current accuracy”.

Here I would like offer some comments as follows

Major comments:

Comment 1: Please add a flow chart of experimental steps in Methodology section (consist of image processing, results evaluation and accuracy assessment).

Response 1: The flow chart is now a part of a manuscript. Thank you.

Comment 2: In Table 1, could you add mean of perimeter estimated from terrestrial SfM at each height level of four tree species?

Response 2: The column for overall RMSE for height level was added.

Comment 3: From line 208 to line 217, the paired t-test was used for evaluating both results, please provide more detail of each tree (t=?, df=?, 95% confidence interval=?).

Response 3: The detailed results from t-test are now submitted as S1 appendix and cited in the section. 

Comment 4: From line 305 to line 309. Could you show clearly a suitable or unsuitable part if the terrestrial SfM photogrammetry were applied in Fig. 6. Also, please make a legend of blue points and red area in Fig.6.

Response 4: The figure was edited. The colour represent kernel density level. The colour was changed and we have added a line based on RMSE of perimeter estimation. The section was edited accordingly. Thank you

Comment 5: Please add conclusion section to reveal that the results achieved from terrestrial SfM photogrammetry can or NOT be used for monitoring the annual tree increments based on accuracy assessment. And could the terrestrial SfM photogrammetry technique is replaced by the tape measurement?

Response 5: Conclusion is now part of the manuscript. The question raised has been answered. Thank you.

Minor comments:

Comment 6: Line 260. Term “reconstruct of trees” can be replaced by “construct 3D model of trees”.

Response 6: Changed.

Comment 7: Line 267. Term “worst RMSE” can be replaced by “highest RMSE”.

Response 7: Changed.

Comment 8: From line 273 to line 287. Please move to introduction section.

Response 8: Section was moved.

Comment 9: I am not an English native speaker but I think the manuscript should be checked by an English native speaker.

Response 9: We have sent the manuscript to English editing service. Certificate is also submitted.

 

Reviewer 2

Dear Reviewer 2,

thank you very much for your comments. In following section, we have answered all your comments and edited our manuscript based on them.

The paper presents an application of structure from motion photogrammetry for measuring trunk diameter increments. This work is novel as it is the first example of measuring trunk increments using a photogrammetric approach and is likely to be of interest to the readers of Plos One.

The paper requires significant improvement in the writing and general presentation before being published. A number of sections are difficult to interpret. For example, the word distract is used instead of subtract in line 110, and units are missing throughout. Nevertheless, the novelty of the work and methods used appear to be sound as such I recommend minor revisions.

Response: 

As we have mentioned above the we sent manuscript to English editing. Furthermore the workflow of whole methodology was added to make it more clear.

Other comments,

Comment 1: Some of the methods are not fully described, for example,

- line 146 "To calculate initial diameter and position of the tree we used the circle fitting algorithm [15]." The circle fitting method is not outlined and reference 15 is a comparison of multiple methods.

Response 1: Clarification has been added. The section was rewritten to bring more light to the workflow.

Comment 2: - line 153, how was the diameter derived from the polygon?

Response 2: We have recalculated the diameter from perimeter. The clarification was added within the section. Thank you.

Comment 3: Figure 2 should provide a scale bar for each point cloud

Response 3: Scale for each point cloud for all three axis was added. 

Comment 4: The results often repeat the methods

Response 4: The results section was edited in a way to decrease the repetition.

---

## [Decision Letter · Decision Letter 1]

21 Feb 2020

Non-destructive monitoring of annual trunk increments by terrestrial structure from motion photogrammetry

PONE-D-19-29708R1

Dear Dr. Mokros,

We are pleased to inform you that your manuscript has been judged scientifically suitable for publication and will be formally accepted for publication once it complies with all outstanding technical requirements.

With kind regards,

John Toland Van Stan II, Ph.D.

Academic Editor

PLOS ONE

Additional Editor Comments:

The revised manuscript has addressed reviewer concerns. In particular, the revisions clarified the methodological elements and the overall workflow, and highlighted key findings/limitations throughout (including the addition of a formal conclusions section).

Reviewers' comments:

Reviewer's Responses to Questions

**Comments to the Author**

1. If the authors have adequately addressed your comments raised in a previous round of review and you feel that this manuscript is now acceptable for publication, you may indicate that here to bypass the “Comments to the Author” section, enter your conflict of interest statement in the “Confidential to Editor” section, and submit your "Accept" recommendation.

Reviewer #1: All comments have been addressed

Reviewer #2: All comments have been addressed

2. Is the manuscript technically sound, and do the data support the conclusions?

Reviewer #1: Yes

Reviewer #2: Yes

3. Has the statistical analysis been performed appropriately and rigorously? 

Reviewer #1: Yes

Reviewer #2: Yes

4. Have the authors made all data underlying the findings in their manuscript fully available?

Reviewer #1: Yes

Reviewer #2: Yes

5. Is the manuscript presented in an intelligible fashion and written in standard English?

Reviewer #1: Yes

Reviewer #2: Yes

6. Review Comments to the Author

Reviewer #1: All comments have been addressed. Therefore, the second revised manuscript was accepted for publication in PLoS ONE.

Reviewer #2: The authors have done a good job in addressing my initial concerns. The paper is now suitable for publication.

7. PLOS authors have the option to publish the peer review history of their article (what does this mean?). If published, this will include your full peer review and any attached files.

Reviewer #1: Yes: Nguyen Van Trung

Reviewer #2: No

---

## [Editor Report · Acceptance letter]

26 Feb 2020

PONE-D-19-29708R1 

Non-destructive monitoring of annual trunk increments by terrestrial structure from motion photogrammetry 

Dear Dr. Mokroš:

I am pleased to inform you that your manuscript has been deemed suitable for publication in PLOS ONE. Congratulations! Your manuscript is now with our production department. 

With kind regards,

on behalf of

Dr. John Toland Van Stan II 

Academic Editor

PLOS ONE